**Assessing runoff sensitivity of North American Prairie Pothole Region basins to wetland drainage using a basin classification–based virtual modeling approach**

Christopher. Spence[1*], Zhihua He[2], Kevin R. Shook[2], John W. Pomeroy[2], Colin J. Whitfield[3], Jared D. Wolfe[4]

\* Corresponding author: Christopher Spence (chris.spence@ec.gc.ca)

[1] Environment and Climate Change Canada, Saskatoon, Saskatchewan, Canada

[2] Centre for Hydrology, University of Saskatchewan, Saskatoon, Saskatchewan, Canada

[3] School of Environment and Sustainability, University of Saskatchewan, Saskatoon, Saskatchewan, Canada

[4] Natural Resources Canada, Ottawa, Ontario, Canada

**Abstract**

Wetland drainage has been pervasive in the North American Prairie Pothole Region. There is strong evidence that this drainage increases hydrological connectivity of previously isolated wetlands and, in turn, streamflow response to precipitation. It can be hard to disentangle the role of climate from the influence of wetland drainage in observed streamflow records. In this study, a basin classification-based virtual modelling approach is described that can isolate these effects on runoff regimes. Three knowledge gaps were addressed. First, it was determined that the spatial pattern in which wetlands are drained has little influence on how much the runoff regime was altered. Second, no threshold could be identified below which wetland drainage has no effect on the streamflow regime, with drainage thresholds as low as 10% by area were evaluated. Third, wetter regions were less sensitive to drainage as they tend to be better hydrologically connected even in the absence of drainage. Low flows were the least affected by drainage. During extremely wet years, runoff depths could double as the result of complete wetland removal. Simulated median annual runoff depths were the most responsive, potentially tripling under typical conditions with the high rates of wetland drainage. As storage capacity is removed



from the landscape through wetland drainage, the size of the storage deficit of median years begins to decrease and to converge on those of the extreme wet years. Model simulations of

flood frequency suggest that because of these changes in antecedent conditions, precipitation that once could generate a median event with wetland drainage can generate what would have been a maximum event without wetland drainage. The advantage of the basin classification-based virtual modelling approach employed here is that it simulated a long period that included a wide variety of precipitation and antecedent storage conditions across a diversity of wetland

complexes. This has allowed seemingly disparate results of past research to be put into context and finds that conflicting results are often only because of differences in spatial scale and temporal scope of investigation. A conceptual framework is provided that shows, in general, how annual runoff in different climatic and drainage situations will likely respond to wetland drainage in the Prairie Pothole Region.

**Keywords:** Prairie, basin classification, virtual experiments, wetland drainage, streamflow

## 1. Introduction

Wetlands exhibit a diversity of functions providing ecosystem services that society values.

Wetlands play active roles in buffering precipitation, storing water, attenuating streamflow and reducing the areas contributing to downstream flooding (Godwin and Martin, 1975; Hubbard and Linder, 1986; Bullock and Acreman, 2003; Acreman and Holden, 2013; Haque et al., 2017). They provide habitat for animal species valuable for pest control (i.e., insectivorous beetles and birds), food sources (i.e., waterfowl), and crop pollination (e.g., bees) (Vickruck et al., 2021).

The periodic hydrological isolation and water retention function of wetlands provides value by



allowing nutrients entering wetlands to be processed and reduced before downstream transport

can occur. High nutrient loading in some lakes and streams in the absence of wetland services

has led to more frequent harmful algal blooms (Ali and English, 2019). Surface water and

groundwater often intersect in wetlands, making them important aquifer recharge locations or

sources of surface water in otherwise dry, arid conditions and environments (Hayashi et al.,

2003). Wetlands also exhibit characteristics that make them vulnerable to removal, despite their

value to society. Urbanization has been a cause of wetland loss for centuries across the globe,

particularly in coastal locations (Li et al., 2018). Wetland removal to expand food production in

agricultural landscapes is widespread (Cortus et al., 2009; Golden et al., 2014; van Meter and

Basu, 2015). Riparian wetlands are commonly removed so that shorelines and riverbanks can be

engineered for better access to water bodies. Estimates of global wetland loss range from 30-

87% depending on the methodologies employed and periods of study (Davidson, 2014; Hu et al.,

2017), with very high rates in some regions and historical periods (Li et al., 2018).

The Prairie Pothole Region located in the Great Plains of North America is a globally significant

wetland-dominated region. As the Wisconsian glaciation ended and continental glaciers receded,

ice blocks were left on the landscape; these formed depressions, prairie potholes, where they

melted. The rain shadow created by the Cordillera to the west results in a dry climate with

limited opportunities for fluvial erosion and drainage network development. This and the

undulating topography have resulted in a landscape with a poorly integrated drainage network

populated with numerous depressions that are hydrologically isolated from one another, except

during rare periods of connectivity. The ponds that form in these depressions range from

ephemeral to permanent, and even a single wetland can have substantial variations in ponded



areas between dry and wet conditions (van der Kamp and Hayashi, 2009). During wet periods,

ponds may fill and spill, or fill and merge, creating intermittent surface water connections among

each other and higher order streams (Tiner, 2003; Shaw et al., 2012; Leibowitz et al., 2016).

Surface water storage dynamics are a critical component of Prairie Pothole Region hydrology

(Haque et al 2017).

A distinct suite of hydrological processes (Millar 1971, Poiani and Johnson 1993, Su et al. 2000,

Niemuth et al. 2010, Liu and Schwartz 2011) controls pothole surface water storage dynamics,

resulting in the functional behaviour that makes these features hydrological and biogeochemical

hotspots. Meltwater from snow that drifts into depressions provides an important source of

water for individual wetlands because evaporation from any open water surface and

evapotranspiration from riparian vegetation generally exceeds rainfall (Woo and Rowsell 1993;

Hayashi et al. 1998; van der Kamp and Hayashi, 1998; Fang et al., 2010). Local runoff from

within the pothole's immediate depression is most likely to occur during snowmelt, when the

ground is frozen and evapotranspiration rates are low (Spence, 2007). High unfrozen soil

infiltration rates direct most summer rainfall into the ground where it is subsequently

evapotranspired (Armstrong et al., 2015). Pothole hydrological connections beyond their local

depressional basins vary in time and space, and these connections occur through intertwined

transient but fast surface water pathways and persistent but slow groundwater pathways (Ameli

and Creed, 2017; Ali et al., 2017). Surface outflow from the depression occurs only when the

pond volume exceeds the depression volume. In the subsurface, when the water table is close to

the topographic surface where hydraulic conductivities are exponentially higher than deeper in

the soil profile, shallow groundwater flux can be large enough to sustain water levels in ponds



that prolong surface water connections (Brannen et al., 2015). Whether groundwater recharges, flows through, or discharges at depressions depends on the topographic location (Winter and Rosenberry 1995, 1998; Rosenberry and Winter 1997; LaBaugh et al. 1998). Surface runoff

flowing into a wetland from outside its depression requires surface storage capacity in upslope depressions to be met such that there can be surface hydrological connectivity from upslope areas. This connectivity determines the area that can contribute to runoff into a wetland or from the wetland complex (Shaw et al., 2012, Hayashi et al., 2016; Shook et al., 2021). As antecedent surface storage and the memory of previous hydrological events strongly dictates the timing and

volume of rainfall or snowmelt allowed to flow downstream, there can be non-linear hysteretic relationships between the area contributing runoff from the wetland complex and the volume of water stored in the complex (Shook et al., 2013, 2021) and between the area contributing runoff and the runoff rate (Mengistu and Spence, 2016).

Changes to wetland hydrological connectivity caused by drainage will alter the function of the wetland complex because functions emerge from how uplands and wetlands connect (Cohen et al., 2016). There is strong scientific consensus that wetland drainage should enhance streamflow by removing depression storage capacity from the landscape (Rannie, 1980; Hubbard and Linder, 1986; Miller and Nudds, 1996; Labaugh et al., 1998, Gleason et al., 2007; Dumanski et al. 2015;

Whitfield et al., 2020; Baulch et al., 2021). Specifically, Wilson et al. (2019) showed that Assiniboine River tributaries having intensive drainage showed higher mean runoff ratios. The strength of this enhancement for any single event or specific watershed remains a point of debate. This is because even if wetland distribution, surface storage capacity and structural hydrological connectivity were spatially uniform, which they are not, there are a multitude of



antecedent wetland conditions that interact with meteorological inputs to influence responses.
These are difficult to control experimentally when using observed meteorological and
streamflow records (Ehsanzadeh et al., 2012). It is very difficult to measure or infer contributing
area, especially for historical events. Using the Cold Regions Hydrological Model (CRHM) to
control for variable state conditions and parameters, Pomeroy et al. (2014) may be the only study
to have been able to estimate the sensitivity of streamflow and found that in Smith Creek, a 400
km$^2$ catchment in east central Saskatchewan, when wetland area was reduced both high and low
magnitude discharge events increased substantially. Wetland drainage was shown to have a
strong impact on floods due to both snowmelt and rainfall. They found complete drainage of
current wetlands resulted in simulated increases of 32% in the annual flow volume and 78% in
peak daily discharge for the flood of record. Conversely, a scenario of restoring wetlands to the
distribution in place in 1958 decreased annual streamflow volume by 29% and the peak
discharge of the flood of record by 32%.

Outside of Smith Creek, the precise response of flood regimes to climate and drainage has so far
eluded researchers and water managers. This is an important knowledge gap because wetland
drainage in the Prairie Pothole Region has been extensive; with removal of between 40 and 71%
of historic wetlands across much of the region, and as large as 95% in the southern edges of the
region in Iowa and Minnesota (van Meter and Basu, 2015). Most wetlands removed during
colonization by Europeans, Canadians and Americans in the late 1800s were small wetlands that
flood temporarily as these were easier to convert to annual crop production (Miller et al., 2009).
More recently, there is evidence to suggest that drainage is focused on 'nuisance' wetlands close
to catchment outlets and raised road embankments that are logistically easier to drain (Lloyd-





Smith et al., 2020). The objective of this research is to address three key knowledge gaps about the influence of wetland drainage on streamflow regimes in prairie pothole basins. First, does

the relative size and location of the drained wetlands make a difference to the change in runoff response? Second, is there a threshold below which wetland drainage has no effect on the streamflow regime? Finally, third, what is the role of climate? That is, do wetter regions and conditions, which presumably have more numerous or frequent connections, have less sensitivity to drainage, as they tend to operate closer to the storage capacity?


## 2. Methods

### 2.1 Framework of classification-based virtual basin modeling

Spence et al. (2022) introduced a catchment classification–based virtual hydrologic model framework that has proven useful for evaluating the sensitivity of prairie watersheds to climate.

This provides a novel tool with which to disentangle the role of climate and wetland drainage on catchment runoff. A catchment classification–based virtual modelling platform has three main components: (1) a classification analysis to derive virtual basin characteristics; (2) parameterization and evaluation of a hydrological model of the virtual basin and (3) application of the model to evaluate response to multiple scenarios. This approach to catchment

classification, virtual basin set-up, and hydrological model application is described with full details in Spence et al. (2022), but is described briefly below for the current study.

### 2.2 Basin Classification

Wolfe et al. (2019) classified over 4000 small catchments (averaging approximately 100 km$^2$)

across the extent of the Canadian Prairie ecozone from the HydroSHEDs dataset (Lehner and Grill, 2013) into seven broad classes each expected to respond in a hydrologically coherent

BY

manner based on geology, topography, wetland distribution, soils and land cover using a

Hierarchical Classification of Principal Components (HCPC) approach (Figure 1). While the

classification approach here follows that described by Wolfe et al. (2019), the classification used

herein does not include climate (temperature, precipitation), as these are instead used as inputs to

the hydrological model (see Spence et al. (2022)), and so the delineation of the seven classes

differs slightly from that shown in Wolfe et al. (2019).

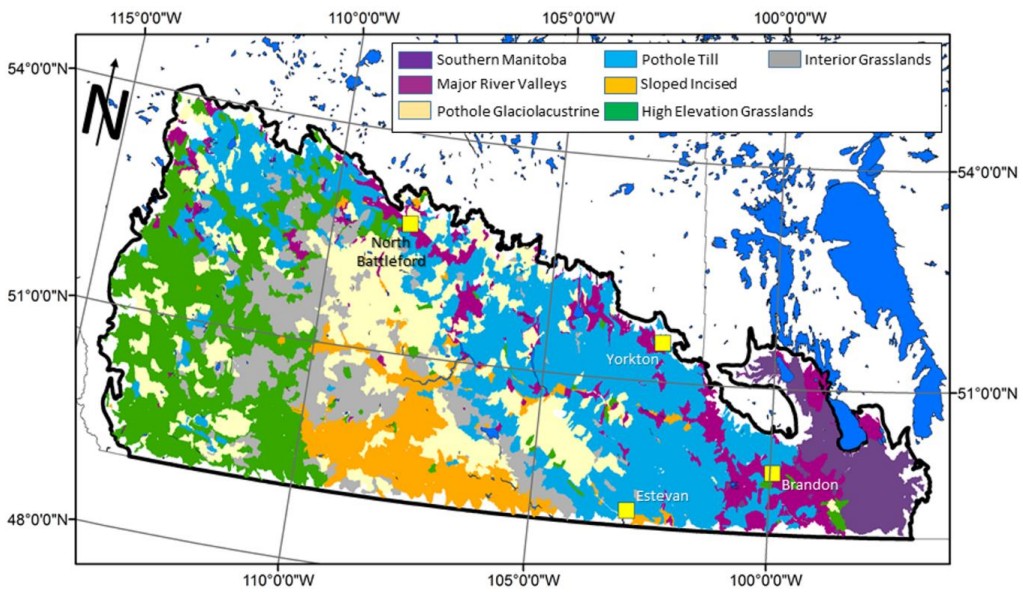

Figure 1: The seven classes of catchments in the Canadian Prairie ecozone. The focus of this
research is on the Pothole Till class (in light blue). Locations of the four climate stations used
are shown (yellow squares).

*2.3 Model set-up and parameterization*



The model application follows that used by Spence et al. (2022) to evaluate the sensitivity of

High Elevation Grasslands (HEG) (Figure 1) hydrology to climate, but instead using the Pothole

Till (PHT) class. The PHT class was selected for use herein, as it has a large geographic extent

(Figure 1), features the highest wetland density and a high cropland coverage (Wolfe et al. 2019),

and is a region of active wetland drainage. This class featured 879 basins for which median basin

characteristics, including area, land cover fractions, basin slope and elevation, soil type, and

wetland distribution were calculated during the catchment classification.

The Cold Regions Hydrological Modelling platform (CRHM) was used to simulate virtual basin

response to four drainage scenarios (see below).  CRHM is a modular, process-based, spatially

semi-distributed hydrological model that includes the key hydrological processes predominant in

western Canada (Pomeroy et al., 2007).  The Prairie Hydrological Model (PHM) configuration of

CRHM (Pomeroy et al., 2010, 2012, 2014) that applies a specific set of modules to represent

Prairie hydrological processes described in Spence et al. (2022) was also applied in this study.

The virtual basin (100 km$^2$) was divided into hydrological response units (HRUs), these are

landscape/drainage areas, each of which has a single set of parameter values informed by the

catchment classification (Table 1).  HRU areas were set according to the median for that land

cover observed across all PHT basins (Table 1). As discussed earlier, wetlands exert significant

control on catchment scale streamflow response.  This control was represented by separating the

virtual basin into non-wetland, and wetland catenas according to median effective and non-

effective fractions of PHT basins, respectively. The first, 'non-wetland' catena routes water

sequentially from cultivated to grassland to shrubland to woodland HRUs and then to the HRU

outlet. The 'wetland' catena portion features a wetland complex which receives runoff from the





other HRUs. Runoff is routed through this complex of 46 wetland HRUs, with the size of

individual wetlands set to follow the shape and scale parameters of a generalized Pareto

distribution determined for the class by Wolfe et al. (2019). This approach has been shown to

effectively represent how wetlands dictate transmission of runoff from Prairie basins (Pomeroy

et al., 2014; Spence et al., 2022). The wetland distribution parameters were derived from

relatively coarse wetland extent data used in the classification by Wolfe et al. (2019). The shape

and scale parameters of this wetland distribution likely underestimate the presence of numerous

small wetlands due to the relatively coarse resolution (minimum wetland pixel size: 30 m by 30

m) available in the remote sensing products.

Drainage scenarios are based on a nominal areal drainage rate (in 10% increments). The result is

that the scenarios can remove fewer wetlands for the same level of drainage than would be the

case if the wetland complex included additional wetlands with smaller area coverage that are not

captured by the remote sensing product. These absent wetlands are too small to individually

influence catchment scale response, unlike a single large depression (Shook et al., 2021).

Accordingly, misrepresenting this part of the wetland area distribution is not expected to bias

model simulations of annual streamflow. However, Shook et al. (2021) showed that using a

coarse representation of a wetland distribution may inflate the role of the largest wetland in

controlling contributing area and runoff, so this should be considered when interpreting these

results.

Table 1: CRHM parameters for the Pothole Till virtual basin model. The suffix "-w" in the HRU
name indicates HRUs in the wetland catena. LAI denotes leaf area index. $D_s$ is the depth to the
lower soil zone (m). When parameters were derived from the literature, references are provided.


| HRU | Fraction of basin | Routing length (m) | LAI (Pomeroy et al., 1999) | Fetch (m) (Pomeroy et al., 2007) | Vegetation height (m) (Pomeroy and Li, 2000) | Manning's n (Fang et al. 2010; Pomeroy et al., 2010) |
|---|---|---|---|---|---|---|
| Channel | 0.01 | 729 | 0.001 | 300 | 0.5 | 0.07 |





| Cultivated | 0.18 | 6449 | 0.001 | 1000 | 0.2 | 0.17 |
|---|---|---|---|---|---|---|
| Cultivated – w | 0.46 | 10688 | 0.001 | 1000 | 0.2 | 0.17 |
| Fallow | 0.01 | 551 | 0.001 | 1000 | 0.01 | 0.05 |
| Fallow – w | 0.01 | 927 | 0.001 | 1000 | 0.01 | 0.05 |
| Grassland | 0.02 | 1988 | 0.001 | 500 | 0.4 | 0.2 |
| Grassland  - w | 0.06 | 3316 | 0.001 | 500 | 0.4 | 0.2 |
| Shrubland | 0.01 | 928 | 0.001 | 300 | 1.5 | 0.2 |
| Shrubland  - w | 0.01 | 1533 | 0.001 | 300 | 1.5 | 0.2 |
| Woodland | 0.01 | 1435 | 0.4 | 300 | 6.0 | 0.4 |
| Woodland  - w | 0.03 | 2371 | 0.4 | 300 | 6.0 | 0.4 |
| Wetland | 0.21 | 97 | 0.001 | 300 | 1.5 | 0.2 |
| | | | | | | |
| Albedo (Armstrong et al., 2008; Male and Gray, 1981) | | | | | | |
| Bare ground | 0.16 | | | | | |
| Snow | 0.85 | | | | | |
| | | | | | | |
| Wetland distribution parameters | | | | | | |
| Shape | 0.87 | | | | | |
| Scale | 2227 | | | | | |
| | | | | | | |
| $D_s$ (m) (Brannen et al., 2015) | 1.4 | | | | | |

## 2.4 Model application

The virtual basin model was run over a 46-year baseline period (1960–2006) using Adjusted and

Homogenized Canadian Climate Data (AHCCD) daily precipitation data (Mekis and Vincent,

2011; Vincent et al., 2012) collected at four locations that represent the variation in climate

across the class (Figure 1; Table 2).  This dataset corrects shifts identified due to station

relocation and changes in observing practices and automation.  Other discontinuities are adjusted

in the dataset with multiple linear regression using a penalized maximal t-test and a quantile-

matching algorithm.  For precipitation, corrections are applied to account for wind undercatch,

evaporation, and gauge-specific wetting losses. Snowfall density corrections are derived based

on coincident ruler and Nipher measurements. Trace precipitation is added. The daily

precipitation data were converted to hourly data required by CRHM using linear interpolation

within its Observation module. The other hourly forcing variables (temperature, relative

humidity and wind speed) were taken from Environment and Climate Change Canada

observations for the same four locations.





Table 2: Location and climate characteristics (1981–2010 climate normal) of the four selected
stations located in and near the Pothole Till class. $T_a$ is mean annual temperature and P denotes
mean annual precipitation.

| Location | Latitude | Longitude | $T_a$ (°C) | P (mm) |
|---|---|---|---|---|
| Estevan | 49° 13' N | 102° 58' W | 3.7 | 427 |
| North Battleford | 52° 47' N | 108° 18' W | 2.1 | 374 |
| Yorkton | 51° 16' N | 102° 28' W | 1.9 | 449 |
| Brandon | 49° 51' N | 99° 57' W | 2.2 | 474 |

*2.5 Model validation*

Canadian Prairie storage state variables often have long hydrologic memories so the first five

years of the simulation period were discarded as these were considered of dubious quality. The

remaining 42-year period of simulation (1965–2006) was used to assess model behaviour. The

virtual basin does not reproduce the hydrology of any specific basin, so there is no single basin

from which observations can be used to assess the model performance. Previous studies have

described the application of CRHM to Canadian Prairie basins and its ability to represent the

region's predominant hydrological processes is well established (Fang et al., 2010) and the

virtual basin model approach has been successfully applied and tested in the HEG class (Spence

et al., 2022). Furthermore, the aim of the simulations was not to simulate specific basins in the

region, but to assess the sensitivity of the hydrological regime to different wetland complex

configurations under climates typical for the region. To assess how the model simulated

streamflow, mean monthly discharge depths for the PHT virtual basin were plotted and visually

compared to the seven Water Survey of Canada stations gauging a stream within 100 km of one

of the climate locations (Table 3) and for which the drainage area boundaries are completely

within the PHT class.




Table 3: Sources of observed data for model evaluation. Water Survey of Canada's (WSC) hydrometric data were obtained from the HYDAT database, available at https://collaboration.cmc.ec.gc.ca/cmc/hydrometrics/www/. Effective drainage area is defined

by Godwin and Martin (1975) as the drainage area that contributes streamflow to the gauged location during the median annual flood was also obtained from HYDAT. These are unavailable for many Manitoba catchments, and are likely underestimates for Saskatchewan catchments because of wetland drainage subsequent to the calculation. Pond level data were obtained from the University of Saskatchewan.


| Streamflow – Water Survey of Canada | | | | |
|---|---|---|---|---|
| *Station number* | *Period of record used for validation* | *Associated climate location(s)\** | *Gross drainage area ($km^2$)* | *Effective drainage area ($km^2$)* |
| 05LL009 | 1975–1994 | Brandon | 171 | n/a |
| 05ME003 | 1975–1994 | Brandon | 1100 | n/a |
| 05MG006 | 1975–1994 | Brandon | 43 | n/a |
| 05MG008 | 1975–1994 | Brandon | 362 | n/a |
| 05NF006 | 1975–1994 | Estevan | 748 | 393 |
| 05NF010 | 1975–1994 | Estevan | 348 | 133 |
| 05ME007 | 1975–1994 | Yorkton | 435 | 57.8 |
| **Pond level surveys - St. Denis National Wildlife Area** | | | | |
| *Pond name* | *Period of record used for validation* | *Associated climate location(s)* | *Strahler drainage order* | *Wetland area ($m^2$)* |
| Pond 1 | 1968–2005 | North Battleford | Fourth | 84000 |
| Pond 2 | 1968–2005 | North Battleford | First | 5000 |
| Pond 25 | 1968–2005 | North Battleford | Third | 40000 |
| Pond 109 | 1968–2005 | North Battleford | Second | 6000 |

*There were no WSC gauges meeting the criteria for North Battleford

Maximum annual pond depths have been measured at the St. Denis National Wildlife Area

(NWA) in central Saskatchewan since the late 1960s. These data represent the only known long

term dataset of wetland storage state in the PHT class. Data from four wetlands with the longest

period of record and fewest data gaps were selected for evaluation of the virtual basin results.

These observed wetlands are connected by intermittent streams, and represent locations on first,





second, third and fourth order channels, though these channels are usually dry. The wetlands

range in size from 5000 to 84000 m$^2$ in size (Table 3). These characteristics represent a diversity

of wetland topologies and geometries. The average annual maximum pond depth for these four

stations was compared to the average annual maximum daily depression storage in all 46

simulated wetlands using correlation analysis. These simulated values are not exactly the same

metric as the observations, but can be expected to respond to climate in a comparable manner, if

the model simulations are robust.

*2.6  Drainage scenarios*

Four sets of drainage scenarios (small-to-large, large-to-small, bottom-to-top, top-to-bottom)

were conducted based on a progression first attempted by Pomeroy et al. (2012) for the

Vermilion River Basin. These sets of scenarios were chosen as they were expected to encompass

the full range of basin response to wetland drainage, even though in reality, drainage will follow

a hybrid of these scenarios, according to decision-making by individual landowners. In each set

of scenarios, depression storage was reduced by progressively completely removing the storage

capacity of individual wetland HRUs, according to either wetland size or wetland proximity to

basin outlet. The drainage fraction ranged between 0 (no wetland HRUs removed) and 100% (all

wetland HRUs removed). Between these states, the nominal drainage was in increments of 10%,

the percentage based on the total original wetland HRU area (0% scenario). The term "nominal"

is used to describe the drainage because, as individual HRUs were removed, it was not possible

to remove exact percentages of the total wetland area, and the actual drainage was set to be equal

to or less than the nominal drainage level. As each wetland was drained, its parameter values

were converted to those of the cropland HRUs as cropland conversion is the normal purpose of

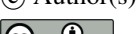



wetland drainage for these agricultural landscapes. These parameters were changed to ensure the

simulation of evapotranspiration, snow redistribution, and soil moisture in the wetland HRU

emulated that of cropland.  In addition, depression storage capacity of the converted HRU was

set to zero and the value of Manning's n was changed to that of the channel HRU based on the

assumption that ditching between the wetlands is associated with wetland drainage.   The

drainage scenarios were designed to indicate the sensitivity of runoff and storage to drainage

when specific parts of the wetland complex are drained (e.g., ones at the bottom of the catena or

ones that are larger) rather than predict the response of any specific drainage scenario that has

occurred in an actual catchment.

Pearson correlation analyses (α = 0.05) were conducted to determine the strength of the

relationship between climate wetness and sensitivity to drainage.  Wetter regions were defined as

those with above average mean annual precipitation or baseline mean annual runoff (i.e., P > 431

mm, Q > 13.5 mm; Yorkton and Brandon). Sensitivity to wetland removal for each drainage

scenario was measured as changes in maximum, median and minimum annual runoff as well as

runoff of different return periods.  The 1:2.33 (median), 1:10 year and 1:42 year return periods

were calculated.  Return periods were calculated with a simple rank technique following Spence

and Mengistu (2019) because Zhang et al (2020) determined that no single frequency distribution

can be used to characterize flood frequencies in the Canadian Prairie, hence the non-typical 1:42

year return period, which is the maximum return period possible for the period of simulation.

Two methods were employed to determine if the sequence of drainage influenced streamflow

response to wetland removal.  First, the simulated values of mean minimum, median and





maximum runoff (for the 42-year simulation period) were collated for the 10, 50 and 90%

nominal wetland removal rates for each of the four drainage scenarios and four climates. The

variation among these scenarios at these drainage rates was used as a metric of the difference

among the scenarios.  Finally, Kolmogrov-Smirnoff tests were run to test if there were

differences among the distributions of annual runoff over the 42-year simulation period (1965–

2006) for each drainage scenario in each climate.  Piecewise linear regression was used to

determine if there was a threshold below which wetland drainage has no effect on streamflow.

The median annual flood and 1:42 year flood simulated for each climate location and nominal

drainage rate were both evaluated for thresholds. Regression was performed in R (R Core Team

2020) using the Segmented package (Muggeo, 2003) which fits regression models with

segmented relationships, and provides, where they exist, the breakpoints between segments.

These breakpoints, where identified, were values at which the rate of change in runoff with

wetland drainage occurs changed significantly.  Thresholds below which wetland drainage has

no effect on streamflow were identified as when the rate of change increased from zero.

In addition to the removal of storage capacity, watersheds with high rates of wetland removal are

more efficient at moving water to the outlet.  In each drainage scenario, one wetland remains as

the last to be removed.  As an indication of how quickly runoff leaves this wetland within each

drainage scenario, the recession coefficient defined by Dingman (1973) was determined from the

wetland storage time series:

$$S_t = S_0 \cdot exp\left(-t/t_*\right) \tag{1}$$

where $S_0$ is defined as antecedent storage, $S_t$ is storage on day $t$, and the recession coefficient $t*$

(days) can be the reciprocal of the slope of the best fit line between $\ln(S)$ and $t$ as storage



declines (McNamara et al., 1998).  Finally, the range and variability of the runoff regime was

calculated for each drainage pattern scenario for each climate using the coefficient of variation of

runoff.

The importance of exceeding depression storage capacity on this landscape for hydrological

connectivity and runoff response has been known since the 1950s (Stichling and Blackwell,

1957).  Leibowitz and Vining (2003) identified that the extent of hydrological connections

should be a function of precipitation, $P$, and local relief, $r$.  The former dictates the supply of

water.  The latter, the capacity with which a catchment can transmit it. The number of

connections, $C$, should be inversely proportional to relief and proportional to the precipitation:

$$C = f(P, 1/r) \tag{2}$$

The Leibowitz and Vining scheme provides a quantitative framework that was applied to

evaluate the role of drainage in enhancing annual runoff.  If fractional drainage, $d$, reduces

effective relief by removing storage capacity and enhancing the ability of a catchment to transmit

water it can have an inverse relationship to local relief. To determine the relationship fractional

drainage and precipitation have with the number of hydrological connections, represented by the

annual maximum connected area, $A_c$, the strength of the relationship was evaluated using

multiple linear regression in R using the lm package (Wilkinson and Rogers, 1973; Chambers,

1992).

$$A_c = f(P, d) \tag{3}$$

Similarly, multiple linear regression was applied to determine if the relationship among mean

annual runoff, $Q$, $d$ and $P$ was like that of $A_c$.

$$Q = f(P, d) \tag{4}$$



## 3. Results

### 3.1 Model validation

The time series of observed pond depths at the St. Denis NWA and simulated depression storage

indicates the virtual model can capture long term behaviour of storage in PHT basins. The

values were reasonably correlated (r = 0.41) as indicated by their time series (Figure 2). The

model runs selected for this comparison were forced with the North Battleford climate, the

closest of the selected climate datasets. Some differences between the model and observation

values can be attributed to the distance and different precipitation inputs between the two sites,

and this is most apparent in the early 1990s. Through 1989 and 1990, St. Denis experienced 42

mm less precipitation than North Battleford. Similarly, St. Denis was 80 mm drier than North

Battleford in 1993. The observational record becomes sparser after 1993, but the model still

tends to capture year-to-year variation in storage and within the same relative amounts as earlier

in the record. The virtual model was able to capture the range of annual runoff observed at the

Water Survey of Canada stations (Figure 3). There was no gauged basin close enough in

proximity to North Battleford that was entirely in PHT, thus this station is absent from the

validation. Runoff regimes among gauged streams close to Brandon can be diverse. The model

simulations captured this range, but not necessarily for a specific watershed, though the virtual

model boxplot was very similar to the runoff behaviour observed for 05ME0003 (Birdtail Creek

near Birtle). Simulated mean annual runoff was overestimated when the model was forced with a

climate from Estevan, but extreme dry and wet years were comparable. Simulations compared

best with observations when forced with a Yorkton climate. Better agreement for some of the





WSC gauges can be expected where those basins exhibit characteristics that are most similar to

the median of all basins in the PHT class used to parameterize CRHM.


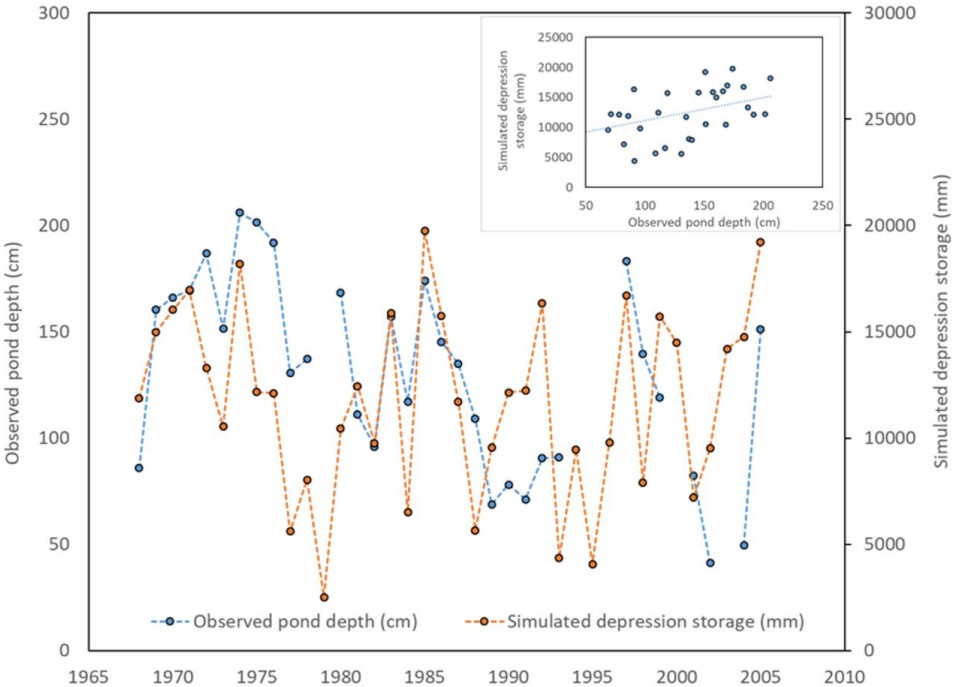

Figure 2: Annual average of observed depths in four ponds at the St. Denis National Wildlife
Area (blue line) and simulated depression storage in the virtual model for the 1968–2005 period
(orange line). The inset illustrates the relationship between the two annual averages ($r = 0.42$, $p = 0.02$).




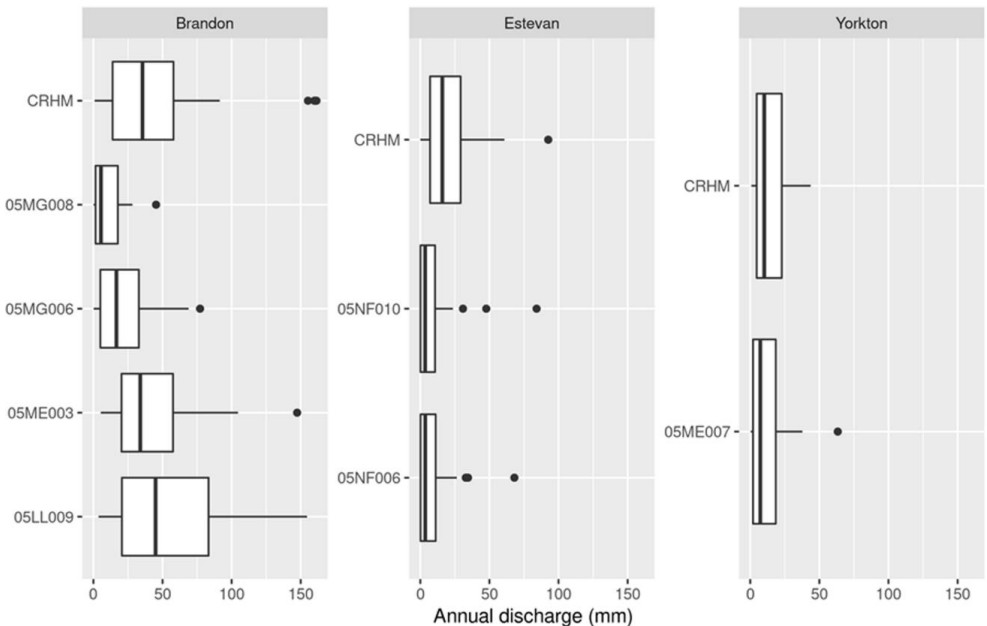

Figure 3: Simulated (CRHM) baseline (recently observed wetland conditions) annual runoff for the PHT virtual basin for three of the four climate locations as well as observed annual runoff from corresponding Water Survey of Canada hydrometric gauges. The vertical line in the middle of the box denotes the mean, and the top and bottom of the box denote plus or minus one standard deviation. The whiskers denote $10^{th}$ and $90^{th}$ percentiles. Circles represent values beyond these percentiles.

*3.2 Role of drainage pattern*

The relatively low standard deviations among the various drainage patterns using meteorological

data forcings for several climate stations (Table 4) show that any difference in the increase in

annual runoff for the same drainage amount was subtle among the four drainage pattern

scenarios. The probability distributions of annual runoff were not statistically significantly

different among the different drainage pattern scenarios at any of the four climate locations.





Kolmogrov-Smirnoff scores were always near 0.1 with p values of no less than 0.8 for the 10%,

50% and 90% nominal drainage rate scenarios for each of the four climates. When 50% of the

wetland area was removed, median annual runoff did not vary much among the four drainage

pattern scenarios, with coefficients of variation of 0.03 (Brandon), 0.05 (Estevan), 0.06 (North

Battleford), 0.07 (Yorkton).

Table 4: Simulated annual runoff (mm) for each drainage scenario for each climate forcing for
the period of simulation. The baseline average for each climate station is provided, as is the
average and standard deviation for each set of drainage scenarios at each of the four climate
stations.

| | Minimum annual runoff | | | Median annual runoff | | | Maximum annual runoff | | |
|---|---|---|---|---|---|---|---|---|---|
| **Brandon (baseline)** | 0.3 | | | 14 | | | 56 | | |
| | **10%** | **50%** | **90%** | **10%** | **50%** | **90%** | **10%** | **50%** | **90%** |
| Small to large | 0.3 | 2.1 | 3.0 | 19 | 28 | 34 | 63 | 77 | 94 |
| Large to small | 0.4 | 1.1 | 3.4 | 19 | 30 | 42 | 68 | 86 | 108 |
| Top to bottom | 0.3 | 0.5 | 1.3 | 19 | 29 | 44 | 63 | 87 | 110 |
| Bottom to top | 0.8 | 2.1 | 3.2 | 18 | 28 | 38 | 58 | 79 | 103 |
| | | | | | | | | | |
| Average | 0.4 | 1.4 | 2.7 | 19 | 29 | 39 | 63 | 82 | 104 |
| St. dev. | 0.2 | 0.8 | 1.0 | 0.2 | 1 | 5 | 3 | 5 | 9 |
| | | | | | | | | | |
| **Estevan (baseline)** | 0.01 | | | 9 | | | 46 | | |
| | **10%** | **50%** | **90%** | **10%** | **50%** | **90%** | **10%** | **50%** | **90%** |
| Small to large | 0.01 | 2.1 | 4.0 | 15 | 21 | 27 | 57 | 67 | 80 |
| Large to small | 0.01 | 0.01 | 2.0 | 12 | 22 | 30 | 59 | 80 | 93 |
| Top to bottom | 0.01 | 0.01 | 0.01 | 12 | 20 | 30 | 56 | 75 | 96 |
| Bottom to top | 0.6 | 2.5 | 4.5 | 14 | 25 | 30 | 52 | 75 | 87 |
| | | | | | | | | | |
| Average | 0.14 | 1.2 | 2.6 | 13 | 22 | 29 | 56 | 74 | 89 |
| St. dev. | 0.28 | 1.3 | 2.1 | 1.6 | 1.2 | 1.7 | 1.9 | 6.5 | 8.8 |
| | | | | | | | | | |
| **Yorkton (baseline)** | 0.8 | | | 14 | | | 44 | | |
| | **10%** | **50%** | **90%** | **10%** | **50%** | **90%** | **10%** | **50%** | **90%** |
| Small to large | 1.2 | 1.6 | 0.8 | 19 | 27 | 34 | 51 | 65 | 85 |
| Large to small | 0.9 | 2.1 | 3.2 | 18 | 30 | 40 | 54 | 72 | 101 |
| Top to bottom | 1.0 | 1.5 | 7.4 | 18 | 26 | 43 | 50 | 67 | 103 |
| Bottom to top | 0.9 | 1.0 | 0.8 | 15 | 27 | 38 | 47 | 69 | 94 |
| | | | | | | | | | |
| Average | 1.0 | 1.5 | 3.1 | 17 | 27 | 39 | 50 | 68 | 95 |
| St. dev. | 0.2 | 0.5 | 3.1 | 0.8 | 2.0 | 4.6 | 1.9 | 3.4 | 10 |
| | | | | | | | | | |
| **North Battleford (baseline)** | 0.7 | | | 7 | | | 41 | | |
| | **10%** | **50%** | **90%** | **10%** | **50%** | **90%** | **10%** | **50%** | **90%** |
| Small to large | 1.0 | 4.6 | 7.1 | 11 | 18 | 23 | 61 | 84 | 110 |
| Large to small | 0.9 | 2.5 | 6.1 | 7.3 | 16 | 27 | 57 | 88 | 132 |
| Top to bottom | 0.9 | 1.5 | 4.8 | 8.2 | 16 | 28 | 58 | 90 | 133 |
| Bottom to top | 2.1 | 5.0 | 8.0 | 11 | 19 | 26 | 54 | 88 | 122 |
| | | | | | | | | | |
| Average | 1.2 | 3.4 | 6.5 | 9.6 | 17 | 26 | 57 | 87 | 124 |
| St. dev. | 0.6 | 1.7 | 1.4 | 2.2 | 1.3 | 2.3 | 2.1 | 2.8 | 13 |






*3.3 Influence of percentage wetland drainage on annual runoff volume*

Model simulations suggest that there are increases in annual runoff even with relatively low

wetland drainage. Median annual runoff increased between 10 and 19% for drainage scenarios

when only 10% of wetland area was drained (Table 4). In low flow years, annual runoff

response across the four climate forcings and drainage patterns exhibited little absolute change,

only increasing by 2.3 to 5.8 mm (Table 4; Figure 4), even once 90% of wetland area was

drained and converted to cropland. This minimal response is largely because when conditions

are dry (e.g., year 2000; Figure 5), there is little water available and storage deficits are high so

little water proceeds to the basin outlet even if there is a reduction in storage capacity with

wetland removal. Baseline median annual runoff at the four locations averaged 11±3.6 mm.

This increased to averages of 15±4.2 mm, 24±5.3 mm, 33±6.8 mm among the climates and

drainage scenarios at the 10%, 50% and 90% nominal percent drainages from current conditions,

respectively (Table 4). Baseline maximum annual runoff averaged 47±6.5 mm. Removing 10%,

50% and 90% wetland area increased average simulated maximum annual runoff to 57±5.3 mm,

78±8.4 mm, 103±15 mm among the climates and drainage scenarios. Complete drainage of

wetlands and conversion to cropland in the model resulted in a more than doubling of simulated

maximum annual runoff, and more than tripling of median annual runoff.

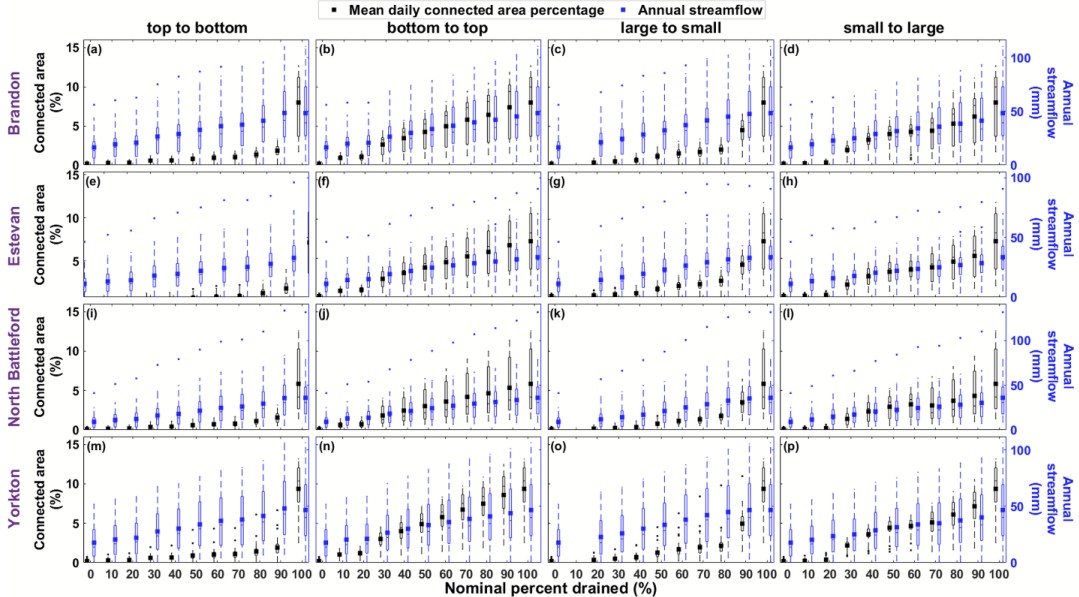

Figure 4: Boxplots of annual discharge depth (blue) and annual maximum connected area percentage (black) for baseline (0% drainage) and four drainage pattern scenarios for each of the four climates for the period of simulation.



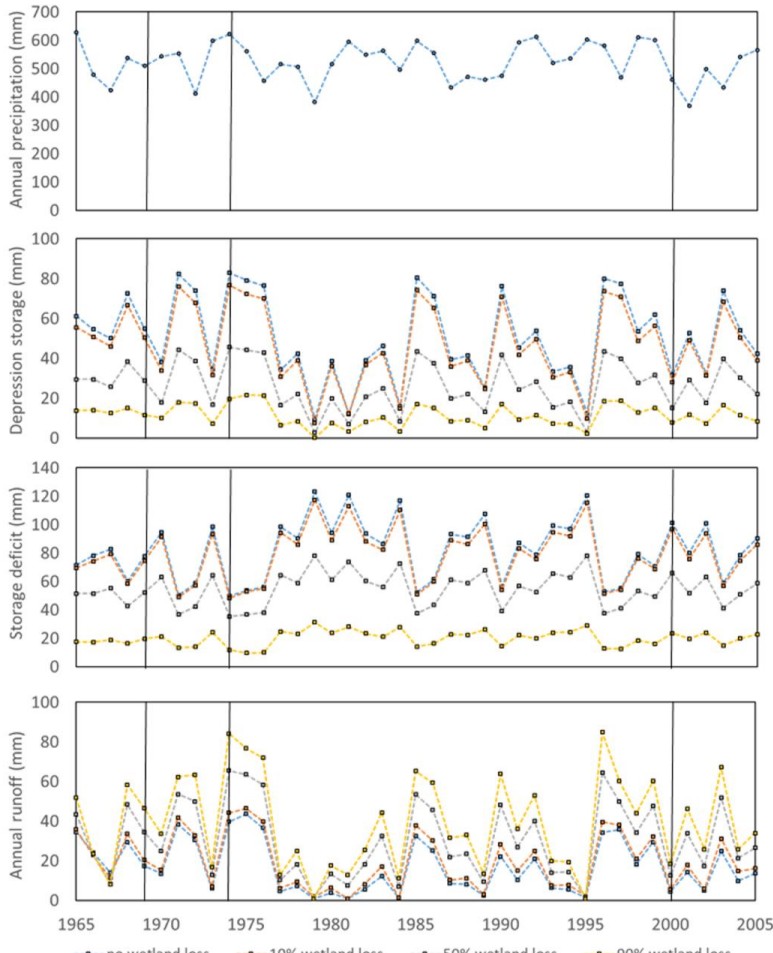

Figure 5: Time series of Yorkton annual precipitation, and simulated depression storage, storage deficit and runoff for the small-to-large drainage scenario. Black lines highlight specific extreme dry (2000) extreme wet (1974) and median (1969) years discussed in the text.

*3.4  The absence of a threshold nominal wetland drainage rate*

No thresholds were found below which removal of wetland storage capacity did not increase

either the median annual flood or less frequent, higher magnitude floods (e.g., 1:42 year flood;

Table 5). Breakpoints did occur in almost every drainage scenario, but these were associated





with a change from a non-zero rate of change in streamflow. The rate increased after the

breakpoint in 16 of the 29 scenarios, and this was almost always associated with the removal of

the largest wetland in the distribution. In the other 13 scenarios, annual runoff continued to

increase with wetland removal, but at a slower rate. This was once, on average, 68% of wetland

area had already been removed.

Table 5: Drainage thresholds (with standard error) where breakpoints indicating a difference in
the rate of change in annual runoff were detected, for two return periods. * denotes a slower
change rate after the breakpoint. Units are in percentages.

|  | Yorkton | North Battleford | Estevan | Brandon |
|---|---|---|---|---|
| **Small to large** |  |  |  |  |
| 1:42 | 80±5 | 80±6 | 80±8 | 81±4 |
| Median | 71±6 | 81±9 | 16±8 | n/a |
| **Large to small** |  |  |  |  |
| 1:42 | 83±7* | 82±5* | 67±2* | 80±4* |
| Median | n/a | 24±5 | 83±6* | 79±5* |
| **Top to bottom** |  |  |  |  |
| 1:42 | 50±27 | n/a | 57±12 | 39±15* |
| Median | 50±19 | 66±4 | 40±12 | 85±5* |
| **Bottom to top** |  |  |  |  |
| 1:42 | 17±1 | 73±11* | 50±3* | 57±7 |
| Median | 30±5 | 78±15* | 49±8* | 50±12* |

*3.5 Role of climate*

Simulated baseline mean annual runoff was used as an indicator of climate wetness as

precipitation alone does not account for the role of evapotranspiration in dictating water

available for runoff. Those locations with drier climates tended to be more sensitive to wetland

drainage but only during extreme years (Figure 6). The change in mean minimum annual runoff

(across the four drainage patterns) over the period of simulation with 100% wetland drainage

was most pronounced with a climate such as North Battleford's, which had the lowest baseline

mean annual runoff. This change diminished with sites that were progressively wetter. Change

in mean median annual runoff was not sensitive to baseline climate. The change in mean annual





maximum annual runoff had a similar pattern to that of low flows but there was less change in

the wetter climates at Yorkton and Brandon. Within a specific climate, however, dry years were

less affected by drainage than median or extreme wet years. Again, using the Yorkton small-to-

large drainage scenario as an example, Figure 5 indicates that dry years were not as influenced

by drainage as wet years with higher baseline flows. This is primarily because in dry years there

is little water available for runoff production, depression storage is low (Figure 5), capacity to

retain storage is high, and areas hydrologically connected to the outlet are small (Figure 4).

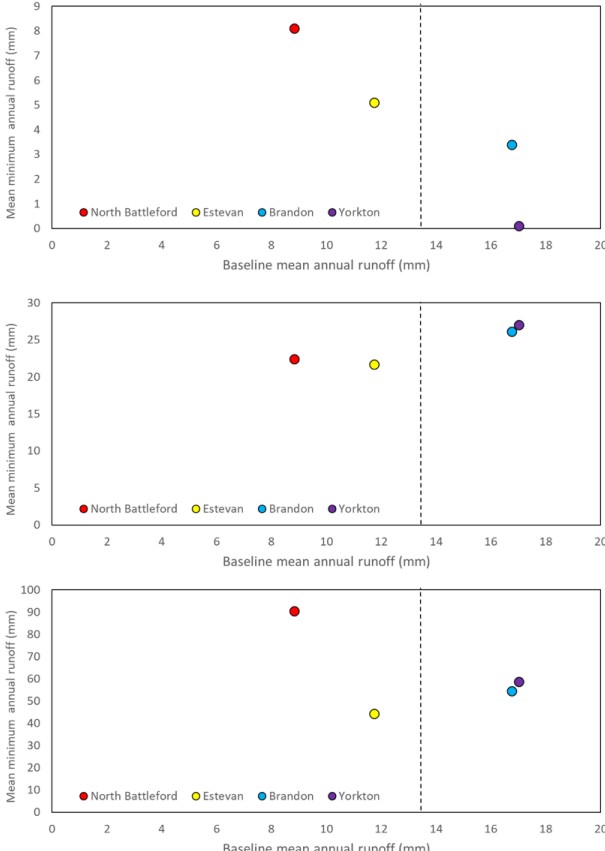

Figure 6: Change in mean minimum, median and maximum annual runoff for each of the four
climate locations compared to the baseline mean annual runoff. The dashed lines represent 13.5
mm/a of median runoff, the average of the four locations.



Catchment scale storage capacities decrease substantially with the drainage of wetlands (from

125 mm to 81 mm with 50% wetland loss). The virtual basin with 50% wetland loss in an

extreme wet year such as 1974 holds a comparable amount of water in depression storage as a

median year without any drainage (i.e., 1969; Figure 5). Annual precipitation in 1969 and 1974

were 510 and 622 mm, respectively. The storage deficit with no drainage in 1969 was 77 mm,

but decreased to 52 mm with 50% drainage, almost identical to that in 1974 (50 mm). With this

smaller storage deficit created by removing half the wetland storage capacity, simulated annual

runoff in median conditions of 1969 doubled from 17 to 34 mm, and converged on the high

water 1974 annual runoff of 40 mm. Likewise, a median year with 90% wetland loss only

retains the same amount of water on the landscape as that in an extreme dry year (1969 vs. 2000,

Figure 5).


As drainage rates increase, the last remaining wetland in each drainage scenario receives more

water from upslope as storage capacity above it is removed. It stays fuller and connected to

downstream locations longer (Figure 7), meaning that separate portions of the catchment can

remain connected to the catchment outlet longer for a smaller amount of annual precipitation.

The storage recession coefficient t* for this wetland for a median year for the baseline Yorkton

scenario was 69 days. For the small-to-large scenario the increase is steady, to 92 days once

10% of wetlands are removed, and 117 and 156 days for the 50% and 90% wetland removal

scenarios. This enhanced hydrological connectivity is exemplified in the top to bottom 90%

drainage pattern scenario. In this scenario there remain relatively small wetlands at the bottom


of a hydrologically well-connected system.  This resulted in the largest annual runoff response to

drainage of any of the four scenarios (Figure 4, Table 4).

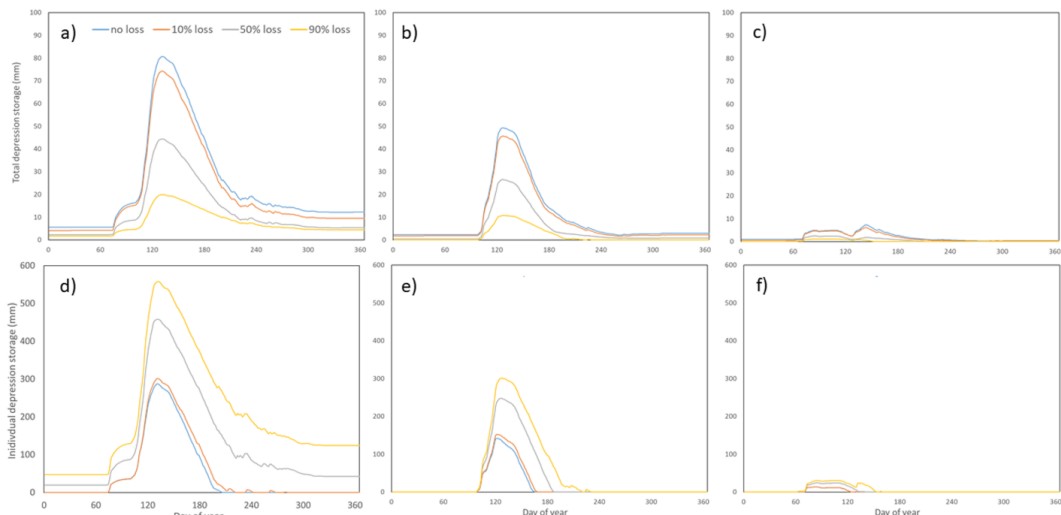

Figure 7:  Total depression storage in mm across the entire 100 km$^2$ virtual basin for a) an
extreme wet year (e.g., 1974), b) the median year (e.g., 1969) and c) an extreme dry year (e.g.,
2000).  Similarly, individual depression storage held in one 0.08 km$^2$ wetland during d) an
extreme wet year (e.g., 1974) e) the median year (e.g., 1969) and f) an extreme dry year (e.g.,
2000).

Sometimes what differentiates a year of extreme high runoff from a median year is high

catchment antecedent storage.  Antecedent depression storage in a median year can be as little as

half that of an extreme wet year (1969 vs 1974; Figure 5), meaning more incoming precipitation

goes into storage in a median year. In addition, because precipitation can be ~100 mm lower in

the median year than a particularly wet year, a larger portion is directed to the storage deficit.

This further suppresses runoff in a median year relative to an extremely wet year. As storage

capacity is removed from the landscape through wetland drainage, the size of the storage deficit

of median years begins to decrease and to converge on those of extreme wet years.  This is why

median year runoff increases faster than runoff in extreme wet years. Model simulations of flood

frequency show that with wetland drainage the same amount of precipitation can generate a

maximum event that would have only generated a median event without wetland drainage

(Figure 8). One key characteristic of all 16 model simulations is the increasing range between

extreme high and low runoff with wetland drainage (Figure 6). The range in annual runoff

increased by 57±17 mm and the coefficient of variation increased accordingly across the four

climates (Table 6).

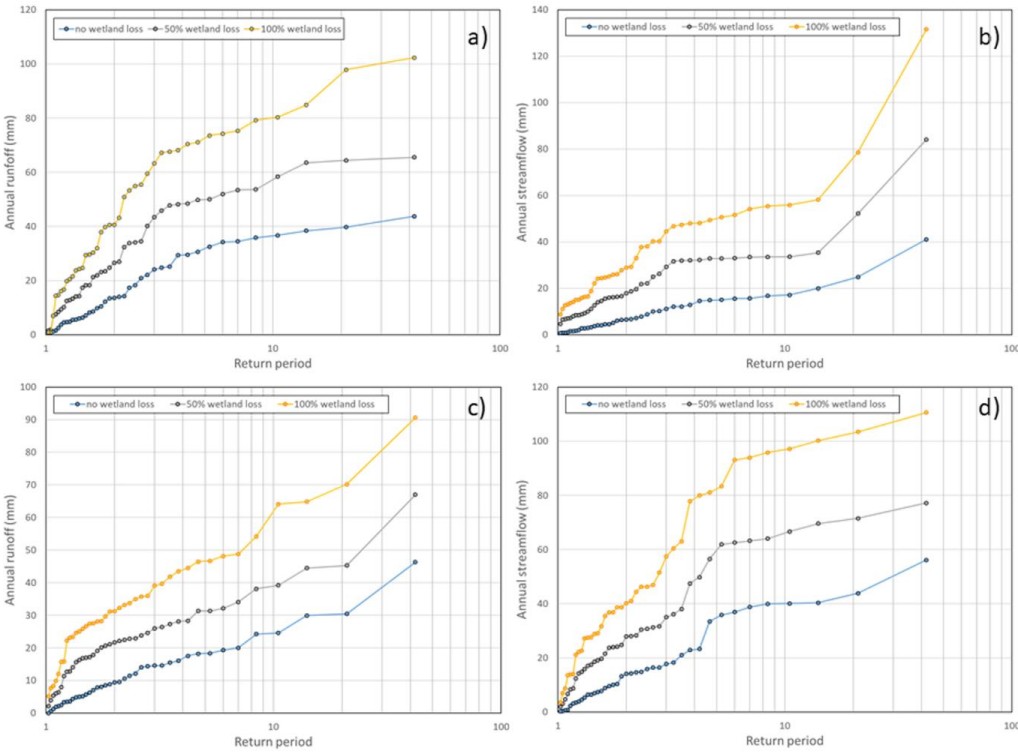

Figure 8: Flood frequency curves for the four climate locations a) Yorkton, b) North Battleford,
c) Estevan, d) Brandon for no wetland loss, 50% wetland loss by area and complete wetland loss
using the small-to-large wetland drainage scenario.



Table 6: Coefficient of variation of annual runoff for the period of simulation (1965–2006)

| Climate | No wetland loss | 100% wetland loss |
|---|---|---|
| Brandon | 0.98 | 1.32 |
| Estevan | 0.99 | 1.75 |
| Yorkton | 1.05 | 1.5 |
| North Battleford | 0.82 | 1.27 |

Applying multiple linear regression analysis to the Leibowitz and Vining scheme indicates that
hydrological connectivity and mean annual runoff increase with both precipitation and fractional

drainage:

$$A_c = 0.01 \cdot P + 0.21 \cdot d - 2.73 \tag{5}$$

$$Q = 0.06 \cdot P + 0.26 \cdot d - 12 \tag{6}$$

Both equations exhibit a relationship with an $r^2$ of 0.9 and p < .001 at a confidence interval of
95% (Figure 9).  Equations 5 and 6 demonstrate simply the positive relationship drainage has

with hydrological connectivity and runoff amount.  The values of the coefficients suggest that
mean annual runoff changes faster with changes in wetland drainage than annual precipitation.

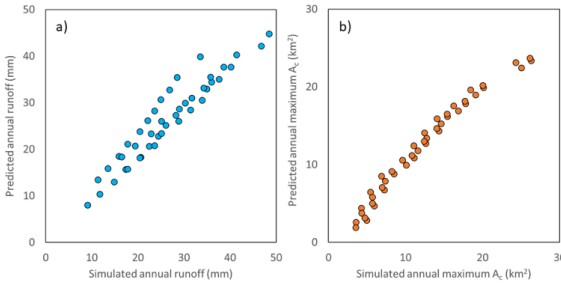

Figure 9:  a) CRHM simulated mean annual runoff vs. mean annual runoff predicted using Eq. 6
and b) CRHM simulated mean annual maximum contributing area vs. mean annual maximum
contributing area using Eq. 5.





## 4. Discussion

The threshold at which runoff responds to drainage is as low as 10% of wetland area (the lowest

scenario evaluated herein), and possibly lower. While current knowledge would imply the

drainage pattern would result in significant differences in the degree of change in annual runoff

(Acreman and Holden, 2013; Shook et al., 2013), it did not. There were no significant statistical

differences in the distribution of simulated annual runoff amongst preferentially draining

large/small wetlands or up-basin/down-basin. While the response to individual runoff producing

events can be different among wetland complexes, model simulations imply when the periods

spanning multiple decades are evaluated, drainage pattern does not make a statistically

significant difference. Upon closer inspection, there were subtle differences in runoff response to

the drainage pattern. When wetlands with larger storage capacity were preferentially removed,

there were shifts in hydrological connections. It is through the influence of wetland storage

capacity on hydrological connectivity that wetland removal influences runoff response. Removal

of 'gatekeepers' (Phillips et al., 2011) enhances the rate of increase in runoff (Table 6) because it

enhances the ability of the catchment to sustain hydrological connections when water is available

(Figure 7). Declines in the rate of increase in runoff sometimes occur at higher nominal drainage

rates. Once all the gatekeepers have been removed, there is less marginal impact of removing

remaining wetlands. These results corroborate those of Shaw et al. (2012) and Pomeroy et al.

(2012) who demonstrated the importance of surface storage state – the volume and the spatial

distribution – to the transfer of water downstream and the dynamics of connectivity across this

landscape.



Higher hydrological connectivity with drainage results in faster rates of filling in remaining

downstream wetlands.  McKenna et al (2019) when applying a model to drained and undrained

cases found that consolidation drainage sped up the rate of filling, and that the earlier filling

caused an order of magnitude more water to spill from the catchment than would have otherwise.

They also found that short pulses of water (priming) are more able to reach the terminal wetland.

This kind of behaviour was documented here with the extended periods of connection in wetland

HRUs under drainage (Figure 7). This explains one reason why large consolidated wetlands in

wetland complexes tend to stay fuller longer than intact smaller ones, because there is a larger

contributing area available to them as the wetlands upslope have been removed (McCauley et al.,

2015). They also tend to be groundwater discharge locations and this augments storage and

keeps them closer to capacity. Because consolidated wetlands tend to be fuller, a wetland

complex dominated by consolidated wetlands is less able to attenuate flooding than the original

complex (Haque et al., 2017).

The major influence of wetland drainage on runoff production is the removal of storage capacity

and resultant increase in basin connectivity.  This allows areas upslope of the wetland that would

have previously only very infrequently contributed to streamflow to become areas that do so

regularly (Tiner, 2003; Ehsanzadeh et al., 2016; Haque et al., 2017), allowing runoff access to

basin outlets.  It is the drainage pattern that most expands the connected area that most enhances

runoff (Figure 4).  While McCauley et al. (2015) claim wetland drainage increases flooding

probability (as also suggested in Figure 8), Hayashi et al. (2016) suggested that wetland drainage

would only impact flood frequency if large terminal wetlands (i.e. gatekeepers) are removed, as

these tend to have the largest storage capacity.  The simulations presented here corroborate both

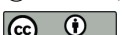



these studies. Even removal of small wetlands increases flooding probability (Figure 8), which concurs with McCauley et al. (2015). Model simulations also support Hayashi et al. (2016) in that they demonstrated that runoff increases faster once large wetlands are removed (Table 5).

An advantage of the virtual basin model approach employed here is that it simulated a long period that included a wide variety of precipitation and antecedent storage conditions across a diversity of wetland complexes.  This has allowed the seemingly disparate results of past research to be put into context and find that conflicting results are often only because of differences in spatial scale and temporal scope of investigation.


For instance, Simonovic and Juliano (2001) concluded that wetland drainage does not enhance low frequency – high magnitude floods, despite removal of storage capacity from the landscape, while Pomeroy et al. (2010; 2014) suggested otherwise. Herein, simulated runoff during both extreme wet and dry conditions was less sensitive to wetland drainage than average conditions.

These similar degrees of change are related to baseline hydrological connectivity.  During wet conditions, baseline connectivity is typically high and storage deficits are low and wetland drainage cannot increase connectivity much more, and runoff response to drainage in extreme years was less than in median years, substantiating Simonovic and Juliano (2001).  During dry conditions, the lack of water results in little hydrological connectivity even with a high rate of

wetland drainage.  With drainage, as capacity is removed, the storage deficit in median years converges on those of wet years, which is why median years increase faster as wetland area drained increases, also substantiating the results of Pomeroy et al. (2010; 2014).





Without capacity on the landscape to hold water, the separation in storage conditions among dry

– median – wet years decreases (Figure 7). Shook et al. (2013) noted how much surface storage

on the landscape instills a degree of memory in the hydrological system. Without surface

storage, precipitation becomes a more important driver of runoff response. In a region such as

the Canadian Prairie where drought and deluge are both common (Johnson et al., 2005) this can

enhance the difference between extremes and reduce buffering of extremes. There are economic

reasons to remove surface water storage capacity from the landscape during wet periods, but

removing the storage capacity of wetland depressions dessicates the region quicker with the

onset of drought.

Model simulations suggest runoff responds to drainage, climate and atmospheric conditions in

five specific ways (Figure 10). The response of the runoff regime to drainage is immediate

during average conditions (Figure 10; #1), but runoff depths increase more if key wetlands with

large storage capacity, gatekeepers, are removed (#2). If drainage continues to 100% loss, the

rate of increase in runoff must slow because the total change has to converge at the same point, if

all else is held equal. However, drier climates experience greater increases in runoff than wetter

ones as there is more capacity to enhance hydrologic connectivity over baseline conditions (#3).

During dry conditions, however, runoff response is tempered as there is little water available for

runoff even though drainage permits greater potential for hydrologic connections (#4). Drainage

enhances runoff during extreme wet years (#5), but not as much as average years, again because

under wetter conditions where hydrologic connections in the basin are greater, the relative degree

of connectivity with drainage does not change as dramatically under a baseline condition.

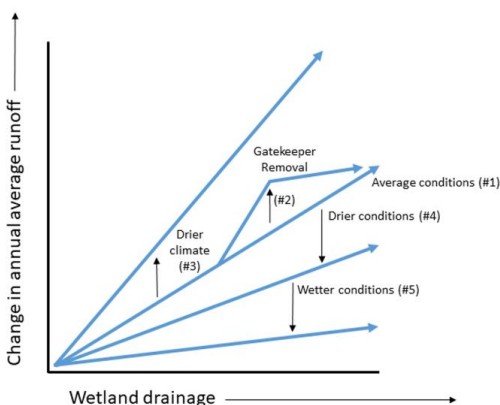

Figure 10: Conceptual framework of runoff response to wetland drainage in the Prairie Pothole
Till class. Five different runoff responses to drainage are shown, representing a range of annual
wetness/dryness conditions and climate. Complete description of these responses is provided in
the discussion text.

## 5. Conclusions

The objective of this research was to address three key knowledge gaps about the influence of
wetland drainage on streamflow regimes. First, is the streamflow response different with the
geometry and topology of drainage (i.e., the spatial patterns of drainage)? Virtual basin model
simulations imply that drainage pattern does not matter to how the long term runoff regime
responds to drainage. However, the removal of gatekeeper wetlands does. Bias that might be
introduced into the wetland distribution by coarse data makes it difficult to identify what the
threshold size of a gatekeeper wetland might be and precisely how much runoff might increase
with its removal. Second, is there a threshold below which wetland drainage has no effect on the
annual runoff regime? No, we were not able to identify a threshold in our analysis, despite



considering drainage levels as low as 10% of wetland area. Third, do wetter regions and conditions, which presumably have more frequent connections, result in reduced sensitivity to drainage, as they tend to operate closer to the storage capacity anyway? Not necessarily. Wetter

climates result in a muted response in extremely high runoff, and climate was inconsequential for median conditions. But in the same climate, minimum, median and maximum annual runoff all reacted differently to simulated drainage. Minimum flows did not change, maximum runoff doubled and median runoff tripled. The removal of storage capacity enhances hydrological connectivity during median annual runoff enough that this runoff approaches maximum runoff

amounts prior to drainage, which results in a faster rate of change than during high runoff conditions when wetlands are well connected and already near their storage capacity.

The response of runoff to wetland drainage is complicated and this has been reflected in the diversity of findings in the literature. The results produced by the virtual basin model imply that

in many instances, these seemingly possibly contradictory results are actually consistent. Differences often have to do with the scope and scale of the study. A conceptual framework and quantifiable relationships are provided that show, in general, how annual runoff in different climatic and drainage situations will likely respond to wetland drainage in the PHT landscape. The authors are working diligently to communicate the information summarized here to water

management agencies, Indigenous peoples, agricultural producers, and watershed stewardship groups. These ongoing efforts will hopefully mean that these results will be used to inform agricultural policies and practices, water management programs and wetland conservation efforts so that everyone in society can benefit from living in this remarkable environment.





**Data Availability**

Climate data inputs, model outputs and model parameter files have been uploaded to the
Federated Research Data Repository. Data are still under the verification of a curator. Once
verified, the data will be publicly available. We expect this to be before the paper is published,
and we already have the doi's that will be used, which we will include in the accepted version of
the paper.

**Author contributions**

CS and CJW conceived the study. ZH, KRS and JWP lead the modelling effort and data analysis.
JDW lead the catchment classification. All authors contributed to writing.

**Competing interests**

The authors declare that they have no conflict of interest.

**Acknowledgments**

Funding provided by the Global Water Futures program to the Prairie Water project. We want to
thank our stakeholders. Financial support over the years from the Canada Research Chair
program, and NSERC Discovery Grants have contributed to the CRHM model development.
We wish to thank Tom Brown of the Centre for Hydrology for his development of CRHM code
over two decades.



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
