# Peer review of "Assessing runoff sensitivity of North American Prairie Pothole Region basins to wetland drainage using a basin classification–based virtual modeling approach"

_Hydrology and Earth System Sciences, 2022_

## Author Comment (AC2)

[Figure]

a)

Cultivated / fallow → Grassland → Shrubland → Woodland

Cultivated / fallow → Grassland → Shrubland → Woodland

Channel

b)

c)

d)

e)

---

## Author Response (AR1)

**Please note that all references to Line locations are those in the clean version of the revised manuscript.**

**Editor comments**

The reviewers raised several important points, one of the most critical being the relationship between this paper and Spence 2022, just published in HESS. The two papers share almost the same title and use the same methodology suggesting significant overlap. The first figure of both papers is also the same. Yet the introduction does not appear to make any mention of Spence 2022. There should be a clear statement in the introduction of what is new in this study compared to the previous one to warrant this additional publication.

We address this comment by adding a clear statement to the Methods section to state the major difference between the two papers. We placed it here as it seemed to make more sense than in the introduction. It is on Line 151: "Spence et al. (2022) successfully introduced a catchment basin classification–based, virtual hydrological model framework that has proven useful for evaluating the sensitivity of prairie basins to stressors. This framework provides a novel tool with which to also disentangle the role of wetland drainage from that of climate on basin runoff. The work presented here is distinctly different from that of Spence et al. (2022) as a different basin class is investigated here as well as a different stressor." In responding to other comments by the reviewers, we have changed Figure 1 and it is now different than in Spence et al. (2022).

**Reviewer #1**

Summary Comment: The topic of the paper is important and timely, however, the methods were extremely hard to follow. There was a lot of important information and clarifying details that were missing from the Methods section. The model calibration and evaluation was weak, and a discussion of the sources of uncertainty in the approach needs to be added.

Thank you for the suggestion. We have made several changes, as the review suggests, the details of which are described below.

Abstract – please add the spatial extent of the study

We added a sentence to address this comment. "The basin class, entitled Pothole Till, which was examined extends throughout much of Canada's portion of the Prairie Pothole Region."

Line 22 – change "were" to "being" evaluated, or revise.

Changed as suggested.

Line 153 – Since Spence et al. 2022 is still in review, please clarify how this effort is distinct from this prior effort. Is the application of the model to a new catchment class the only difference?

Since this manuscript was submitted, Spence et al. (2022) has been accepted and published. We have updated the content in the current paper. Please see our response to the editor above in regards to this comment.

Lines 171-172 – is the data that is shown in Figure 1 an output from Spence et al 2022 or from Wolfe et al 2019 or another source? It is hard to tell.

The classification shown in Figure 1 is not from Wolfe et al. It was another classification that did not include climate as an input that is used in this exercise and was used by Spence et al. (2022). This is

hopefully made clearer by the removal of "(see Spence et al., 2022)". Citing that paper is not necessary here.

Figure 1 – Indicate or clarify what the actual extent of the model and simulated data is? Or clarify in the text that the theoretical model was forced with 4 different climate datasets. If this was the case, what was the size of the simulated basin?"

Thank you for the comment. Much of the information is already in the paper. The caption in Figure states that the focus is on modelling a virtual basin of the Pothole Till class. This is also stated on Line 186 ", but instead using the Pothole Till class." In Section 2.4, *Model application*, we state the virtual basin model was run over a 46 year period using data collected at the four locations shown in Figure 1 and listed in Table 2. On Line 203, we state "The virtual basin (100km$^2$) …"

Lines 183-185 – add some basic information on the land cover, slope, elevation, and soil type, what is the spatial resolution and source of these datasets? Was land cover assumed to be stable or non-changing over the modeling effort, other than changing drained wetlands to agriculture?

We have added some detail at Line 173; "The classification approach here follows that described by Wolfe et al. (2019), using the same elevation (Farr et al., 2007), water extent and distribution (Pekel et al., 2016), surficial geology (GSC, 2014), soils (AAFC, 2015), land use (AAFC, 2016) and tillage practice (Statistics Canada, 2016) data ….."

These data permitted the typical land cover and agricultural practices in the Pothole Till class over the modelling period to be identified. It was this information that was used to parameterize the virtual basin model. Land cover (i.e. crop types, summerfallow fraction) was held constant over time in the simulations apart from converting wetlands to cropland with drainage.

Line 193 – clarify what size HRUs are used.

We have changed the column in Table 1 that listed the HRU fraction of the basin to list HRU area so that these areas are clearer to the reader.

Lines 197-201 – Why not use actual wetland datasets instead of artificially created ones? There are some existing efforts by Amani et al. and Mahdianpari et al. for instance.

This is because the virtual basin modelling approach simulates no specific place within the basin class, only a representation of the typical basin within it. This avoids errors known to exist in many wetland datasets that are used to represent depressional storage in a hydrological model. We have tried to make this point clearer by adding more explanation at Lines 157; "In this framework, a hydrological model of a virtual or stylized basin is parameterized using the predominant characteristics of a class. The model inputs or parameters can be manipulated to simulate the probable response to wetland drainage within a region. The output can be considered representative of how the whole of the basins of that class would respond."

Line 202 – What DEM was used to route runoff? Were any manipulations/changes made to the DEM to condition it hydrologically?

We did not need a DEM, only the catena as described in the paper, and the average routing distances of the land cover types calculated across the Pothole Till class. We have added a sentence; "Routing distances across each HRU were calculated as the average across the 879 basins in the Pothole Till class (Table 1)."

Lines 207 – Please just list the actual source of the wetland extent data, instead of pointing to another publication, which did not generate a wetland dataset from what I can tell.

We provide this water extent data citation now on Line 175. The sentence here is actually referring to the shape and scale parameters, which were determined by Wolfe et al. (2019), so we believe this is the correct reference in this instance.

Lines 211-216 – So no actual data on drainage was used? How did the simulated drainage account for drainage already present? Or was this assumed to not be relevant since the wetlands were simulated as well? It was also difficult to tell what the term "drainage" was referring to. In the U.S. PPR, drainage is usually installed under the ag fields, although historically ditches were also used to drain wetlands. Was drainage simulated by "removing" wetlands or just increasing the rate at which water left the wetland. This is important to clarify.

No. This is for two reasons. First, the virtual model did not represent a specific basin, so an actual real-life drainage plan was not simulated. The reviewer's comment suggest some description of how wetlands are drained in the Canadian Prairie is necessary as it differs from that in the United States portion of the Prairie Pothole Region. We have added a sentence at Line 313 explaining this; "Wetland drainage in this region is typically enacted by first removing any woody vegetation from around the wetland with backhoes and graters. Infilling and levelling is used where possible to flatten the depression. Ditches are dug between each depression to their maximum depth following the local grade to allow drainage towards the closest intermittent streambed or road ditch. These drainage techniques completely remove wetland depression storage capacity from the landscape". For this reason, we did not need to account for drainage already present, as this would be reflected in the baseline wetland distribution that was used. We are wondering if the reviewer is referring to what is called "tile drainage" in Canada. While used in Ontario and parts of southern Manitoba to drain level fields, this is not the practice we wanted to assess and it is rarely used to drain prairie pothole wetlands. The drainage was simulated by "removing" wetlands and their storage capacity and converting that area in the model to the cropland HRU. This did involve decreasing the surface roughness to allow for faster runoff in the routing module. This is explained in Section 2.6, where much of this content has been moved, as upon reading again, seemed a better place for it.

Table 1 – A lot of these parameters are not explained. The routing length is the length from where to where? Is this the average length? The LAI values seem weirdly low…why are the 0.001 in a grassland, cultivated field and shrubland? And the woodland also seems low, a LAI of ~1.5-3 would make more sense here. Look at an example paper to more appropriately parameterize these such as Asner et al. 2003, Global synthesis of leaf area index observations: implications for ecological and remote sensing studies: Global leaf area index. Depending on how the model is set up, this could influence the model findings.

We have better explained routing length in the caption, which is the distance across the HRU to the downstream HRU. The LAI values are low as they are the minimum annual values and typically represent winter where they are used as inputs to blowing snow simulations. This was a mistake and we thank the reviewer for catching this. We have changed them to the annual maximum values, which are more informative.

Line 230 – what is the spatial resolution of the precipitation data, is it station data given that it is collected only at 4 locations?

It is station data collected at four locations representative of the diversity of climate in the basin class. Each set of drainage scenarios was run four times, using the station data representative of different climates in the region.

Line 252 – if 1965-2006 was used to assess model behavior, what years were used to train and calibrate the models? And if actual discharge was used, how did you guys account for the influence of existing land use and wetlands on the discharge values?

We should have better explained CRHM and relied less on referring people to Spence et al. (2022). CRHM is strongly physically based and does not require model calibration. We have added a description of this on Line 266; "The models created with the CRHM algorithms, especially for its surface processes, are strongly physically based, and do not require calibration from streamflow. Furthermore, as a virtual basin has no specific location, it cannot be calibrated to streamflow observations from a gauged basin. As there are few unregulated gauged basins of the size simulated here in the Canadian PPP which is a sparsely gauged region, using a model in which parameters are set based on hydrological process research rather than calibration is advantageous."

We accounted for differences between the virtual basin and the observed discharge values from actual basins by comparing to the range of observed discharge values that occur in the region.

Table 3 – where are these gages? Can these be added to figure 1?

Yes. These have been added.

Line 279-280 – where is the St. Denis NWA in relation to the study area?

This has also been added to Figure 1.

Lines – 286 – I still can't quite tell where you guys ran this simulation – across the entire pothole till? just within or near the catchments in table 3? So I can't tell what the distance is between these ponds and the modeled area, but since the climate stations used are spread across Canada, I have to assume that there are thousands of miles between some of the simulated areas and the ponds, consequently the pond depths are only compared to 1 climate forcing, but this doesn't seem adequate given that the focus of the paper is on changes in simulated wetland extent and corresponding changes in discharge.

We are hoping the new content on Lines 155 better explains that by definition, the virtual basin does not have a real geographic reference. It only represents a typical Pothole Till basin. The climate stations are from the basin class and represent the range of climates that a Pothole Till basin could be exposed to. We have added St. Denis to Figure 1 and compare those data to the runs using a North Battleford climate, as this is the closest station to St. Denis. This is stated on Lines 421. We also added content about how far apart they are; "… the closest (162 km) of the selected climate stations."

Section 2.6 – please clarify how the 4 scenarios differ….so bottom-to-top for example…wetlands were drained sequentially from the basin outlet toward the headwater streams? Was any attention paid to whether the wetland was near- or connected to a stream, or geographically isolated from the stream network in the scenarios?

We have elaborated on this at Lines 318; "Four sets of drainage scenarios (two based on area and two based on relative location) were implemented based on an approach progression first demonstrated by Pomeroy et al. (2012) for the Vermilion River Basin, Alberta. The two scenarios based on area first drained wetlands 1) from smallest to largest and 2) largest to smallest and are referred to as small-to-large and large-to-small, respectively. The two scenarios based on relative location first drained wetlands 1)

from those farthest from the basin outlet to those closest and 2) from those closest to the basin outlet to those farthest.  These are referred to as top-to-bottom and bottom-to-top, respectively."  In the relative location scenarios, yes, these were designed specifically to figure out the response based on those closer to the stream and those more likely to be geographically isolated (e.g., those farthest away).

Line 328 – add a comma between mean and minimum

Fixed.  Thanks.

Line 383 – what was the distance between the N. Battleford climate and the ponds?

162 km.  This detail has been added to the text at Line 422

Lines 395-399 – what figure are these results reflecting?

Figure 3.  We added a reference to it.

Figure 3- Did the 4th location not have gage data? And since the contributing areas were of different sizes, would it make more sense to normalize the x-axis, so annual discharge per contributing area? Otherwise the Brandon comparisons really look all over the place. Also please add these value in as a table so there is some quantitative way to compare.

The 4th location (North Battleford) did not have any gauges close enough to it that met the criteria.  This is noted on Line 429; "There was no gauged basin close enough in proximity to North Battleford that was entirely in PHT, thus this station is absent from the validation."

The annual discharge is normalized; it is expressed in mm in Figure 3.  The numbers have been included in a new Table 4.

Figure 7 – make the font size larger, it was difficult to read this figure.

These have been fixed.

Discussion – the discussion does a nice job of contextualizing the results with the findings of others but please add a paragraph discussing the limitations of the modeling approach and sources of uncertainty given the input datasets, and limited manner that the model performance was evaluated.

We have added some content on Lines 663 (end of third paragraph in Discussion) to address this comment.  "Something to consider with the basin classification based virtual modelling approach is that the results are representative of what would be expected in a typical PHT basin, and not any specific basin.  Because the model does not represent a specific basin, good model performance should be determined not necessarily on how well simulations emulate observations from one place, but how well the variability in hydrological behaviour is captured.  Departures from the modeled results will exist, depending on how different a specific basin is from the parameterized virtual basin.  The results are best interpreted as how basins across the class as a whole would respond to wetland drainage. "

Comment – please revisit how the term "drainage" is used throughout the paper (e.g. line 659) since the term drainage typically refers to the movement of water through a watershed, but here it is mostly used to refer to the action of draining wetlands.

We address this comment by adding a sentence in Section 2.6 – Drainage scenarios that specifically states: "In the context of this paper the term "drainage" refers to wetland drainage; the act of removing surface water storage capacity from depressions, and not the movement of water through a basin."

**Reviewer #2**

Overall, the subject matter, methodological approach, and results of this paper address relevant scientific questions within the scope of this journal. The authors present an interesting modeling application towards better understanding complex landscape hydrological processes.

It is unclear the actual spatial extent of the modeling study. I suggest including more maps in text and in supplemental material to orient the reader through the complex methods. It is clear the Cold Regions Hydrological Modelling platform (CRHM) has been developed and published in many different publications, but it would be helpful to include a spatial visualization of routing pathways of surface water, spatial orientation, and size of wetlands relative to stream networks and how these watershed characteristics change under different drainage scenarios. These details are currently lacking or hard to find, despite the major conclusions of the paper directly relating to these aspects.

We have added a new figure (Figure 2) that includes a spatial visualization of the routing among the HRUs, including the relative size and locations in the virtual wetland complex in the no drainage scenario and a 30% wetland loss scenario. This new figure would have a caption: "Panel a) illustrates the runoff routing among the HRUs showing the non-wetland and wetland catenas, where the latter includes routing runoff from the non-contributing portion of the basin through a wetland complex. The relative sizes and locations of the wetlands in this complex are conveyed by the squares in panel a). Panels b) through e) illustrate an example of the complex under a 30% drainage scenario (the removal of 30% of wetland area) which would be substituted into the wetland catena in panel a). Panel b) is the small to large scenario; c) the large to small; d) the top to bottom and; e) the bottom to top."

[Figure]

Another point of clarity is to standardize use of stream discharge, streamflow depth, and runoff. These different terms are used throughout interchangeably and makes the findings hard to follow.

Yeah. This is our mistake. We have removed all use of the word "discharge" when we really mean "runoff". Furthermore, because streamflow is often expressed as volume per time, and "streamflow depth" is kind of misnomer, we have adopted the term "runoff" throughout as we present all our results as depths in mm.

Overall, interesting conclusions are made but more effort could be made to distill the major findings and move some method and results information into supplemental material.

Thank you for the suggestion. We appreciate the desire to keep a manuscript succinct. We feel it is important to provide all the relevant material in the main body of the paper.

Below are specific editorial suggestions regarding text, tables, and figures.

L63 this half of the sentence does not say anything. Consider removing or re-writing

We feel it important to note that wetland loss is not evenly spread through time or space, so we re-wrote it. "Rates of wetland loss are not the same everywhere, and some regions and periods have experienced very high rates of loss (Li et al., 2018).

L71 replace "numerous depressions" with an order of magnitude (i.e., millions) estimate of number of basins

We replaced this with millions.

L80-109 this paragraph can be distilled and shortened to focus on how this information describes the system and problems

We have shortened and tried to focus it on processes important for assessing response to wetland drainage as suggested.

L110 This sentence is confusing and repetitive, consider re-writing

Fixed.

L143 These objectives are a perfect spot to set the tone on standardizing language in regards to stream response variables of interest (streamflow, runoff, discharge) and then keep consistent after that

Changed to "runoff" throughout, as explained in a response to an earlier comment.

L195-210 this section would be much easier to follow and would allow for better interpretation of results if there were at least one map showing the distribution of wetlands and runoff preferential flowpaths under different drainage scenarios. I find it hard to visualize what this looks like in virtual model space.

Please see our response to the first major comment above.

L293 More details about the drainage scenarios are needed to better understand what is being manipulated in the model relative to the hydrologic responses.

Please see our response to the first major comment above.

L307-315 This text would be easier to understand with a visual figure as well. Could be in the supplemental material

Please see our response to the first major comment above.

Fig 1. The weather station sites are very hard to see. Make bigger and more contrasting to the background

We have changed this, and other aspects of the map in response to Reviewer #1's comments.

Fig 2. Include slope and intercept of regression model in caption. Also, make sure it is noted why the simulated depression storage is an order of magnitude larger than the observed pond depth. Consider changing units on one y-axis so you are comparing mm to mm or cm to cm

It is not the absolute magnitude we are comparing here, but the year-to-year variation in two terms – one modelled and one observed – that represent surface water storage on the landscape. For this reason we feel that changing the units would not necessarily help; it may confuse the situation by implying that the order of magnitude should be the same between the two terms. We have added the slope and intercept of the model in the caption.

Fig 3. This figure does not say that much and could be moved to the supplemental material

We respectfully disagree. Because it is so difficult to evaluate a virtual basin model we feel it is important to have these results front and centre in the paper. Reviewer #1 requested the numbers that make up this figure available in a table. This table has been added.

Table 4. In L416-418 you show that there is low deviation between drainage scenarios. To simplify this information consider moving this whole table to the supplemental material and only present the average and sd for each site

We do not wish to include a supplemental section to the paper so we have reduced this large table as suggested to only present the average and standard deviation for each site.

Figure 4. Similar to Table 4 suggested edits. Take average of all 4 scenarios and only visualize that in the main text. Move whole figure to supplemental. This figure is hard to pull details out of. Also, in the caption include the time period that is modeled and the units of discharge depth. Discharge depth is mentioned in the caption, but the figure shows Annual streamflow (mm).

We admit there is a lot in the figure, which we feel needs to be shown and available in the main text. Because each of the box and whisker plots already includes the variation among the 30 year simulation period, it would be problematic to average the four scenarios into one box plot as well, and retain the information about variability throughout the simulation period. We require the 30 year box and whisker plot to demonstrate a few key characteristics of each drainage scenario that are discussed in the text (e.g., dry years remain dry even if there is lots of wetland drainage). We would like to retain the figure as is, and have updated the caption and axis labels as suggested.

Figure 5. add "median" "wet" and "dry" labels above the top panel next to each corresponding black vertical line.

Done.

Table 5. This information is confusing. Maybe because I do not think about 1:42 as a flood size very often.

Granted it is. To simplify things we have removed the 1:42 year flood from the table. The message remains the same even if only the median flood is discussed, that we could identify a threshold below which the high streamflow of the year was not impacted by any wetland drainage. We have tried to clarify these points in Section 3.4 too.

Figure 6. This could be consolidated into one panel with the bottom panel's y-axis range. Right now the y-axis labels seem to be wrong and missing median and max labels.

Updated as suggested. *

Figure 8. Different y-axis labels. Standardize runoff or streamflow and be consistent

Fixed.

L548 needs a new section title since this analysis and Fig. 9 talk about a different topic.

We respectfully disagree. The content is not quite extensive enough to warrant its own section. The topic does have to do with the role of climate on the response of hydrological connectivity and runoff to wetland drainage.

---

## Author Response (AR2)

Thank you for the good news about the conditional acceptance of the paper.  We summarize the remaining concerns below and provide our responses in red.

I am pleased to accept your paper subject to technical corrections. My final remark is that the font of many figures is too small to be readable by naked eye on a A4 scale.

We looked again at the figures and agree Figures 4, 5, 6 and 9 need larger font sizes.  We have increased the letters in these four figures.

Your reference list includes works "in review". Such works can be cited upon submission if being available to the reviewers. They should not be cited in the final, accepted manuscript, unless published, accepted for publication, or available as preprint with a DOI.

This paper has since been accepted and published.  We have updated the citation.